# Gamasina Mites (Acari: Mesostigmata) Associated with Animal Remains in the Mediterranean Region of Navarra (Northern Spain)

**DOI:** 10.3390/insects10010005

**Published:** 2019-01-05

**Authors:** Sandra Pérez-Martínez, María Lourdes Moraza, Marta Inés Saloña-Bordas

**Affiliations:** 1Department of Environmental Biology, University of Navarra, Pamplona 31008, Navarra, Spain; mlmoraza@unav.es; 2Department of Zoology and Animal Cell Biology, University of Basque Country, Bilbao 48940, Basque Country, Spain; m.salona@ehu.eus

**Keywords:** Acari, Mesostigmata, biodiversity, organic decomposition, Mediterranean region, Navarra, Spain

## Abstract

Mites should not be overlooked as a forensic tool, as many are commonly associated with decomposing animal matter and are closely associated with specific insect carriers and habitats. It is necessary to increase our understanding of the diversity of mites that are found in human and animal remains, their geographical distribution, and their population dynamics. This work is the first study of the role of mites in forensic science in the Mediterranean region of Navarra (northern Spain). Samples were taken using three types of traps (96 modified McPhail, 96 modified pitfall, and 32 carrion on surface) baited with pig carrion during the period between 11 April and 24 June, 2017. Insects were collected in 100% of the traps and only 27% of them contained mites. Information on 26 species of mites belonging to seven families, their ontogenetic phoretic stage/s, their abundance, and presence/absence during the spring season of the study is given. The most abundant species collected were *Macrocheles merdarius*, *Poecilochirus austroasiaticus*, and *Poecilochirus subterraneus*. We are contributing 16 new records for the Iberian Peninsula: seven species of Parasitidae, three species of Macrochelidae, four species of Eviphididae, one species of Halolaelapidae, and one species of Laelapidae.

## 1. Introduction

Mites inhabit nearly every ecosystem throughout the world, although most species are associated with specific environments. Some mites have adapted to ephemeral environments such as carcasses and corpses, which represent a limited food source exploited by other organisms, including their host necrophilous insects with which mites share similar trophic and ecological requirements [1,2,3,4,5,6,7,8,9,10,11]. In such habitats, when the food source modifies or disappears, the altered environment may become unsuitable to its inhabitants. Should this happen, the inhabitants could be forced to migrate to new habitats in which to complete development or reproduction. Mites are small animals devoid of wings and some have acquired a phoretic strategy [8,12] to facilitate their dispersion to other food resources. The phoretic species should assume the risk of losing population density, as some specimens may become detached and perish during the journey; it is also possible that none of the mites arrive successfully to a suitable habitat [6]. Despite these risks, this strategy is advantageous to the phoretic species, not only in terms of its survival, but also in terms of allowing the species to potentially expand its geographical distribution range [13]. The relationship between the host and its phoretic mite may be the result of a co-evolution process. The phoretic travel may involve mite preference in the choice of a host (specificity) and mite selection of the ontogenetic stage or sex to migrate [12,14].

The remains of animals represent dynamic and ephemeral ecosystems, changing over time and eventually being reduced to dry substrate in a relatively short period of time. As a consequence, sarcosaprophagous mites may act as generalists and develop phoretic instars in their life cycles [12,14]. Different species may develop different strategies, adapt different instars to phoresy, and develop special/specific structures to facilitate attachment to a host during the transport [7,9,12]. The first mites will arrive with the first flies (blowflies and flesh flies) and will detach from their hosts as soon as they land on the remains in order to lay eggs. A few hours or days later, additional species will arrive on beetles, and therefore a specific model of succession may be established for mites in parallel to the insect community [7,9].

In ephemeral environments, phoretic mite species could be generalists and appear throughout the year travelling on a variety of hosts, while in long-lasting environments they would depend on their specific carriers.

At least 127 species of mesostigmatid mites have been reported using the phoretic strategy to disperse and several species are predominant on animal and human corpses [7,9]. The families most commonly associated with necrophilous insects are Macrochelidae, Parasitidae, Ascidae, and Uropodidae. The reproductive process of these mites can be by means of anfigonic sexual reproduction and/or parthenogenesis, depending on the families and even the species. Their life cycle includes three immature stages (larva, protonymph, and deutonymph) followed by the molting of each, and finally the adult stage. All stages are active, although the larvae do not feed in some cases (i.e., some phytoseiid species) and some mites do not feed during the phoretic stage in other cases [6]. The phoretic stage (phoront) differs among families not only in the location where they attach themselves to the hosts but also in the manner with which they attach themselves, and in some species a polymorphism even exists between non-phoretic and phoretic individuals [6,15,16,17]. At the same time, it is not uncommon to find polymorphism among the phoretic population [18]. The existence of polymorphic species must be taken into account by forensic entomologists to avoid errors in species identification. Moreover, different phenotypes may have different biology and ecology and therefore provide different forensic information.

Mites associated with animal remains can be considered a potentially useful tool in forensic investigations due to their specificity, abundance, and diversity. They can be found in almost every type of environment with high accuracy in habitat requirements such as stored food, clothing, and decomposing matter [9]. Due to their small size (0.1–1 mm), mites are able to colonize relatively inaccessible places before other animals, may go unnoticed in transported goods, and may cause the introduction of exotic biota. They can be easily overlooked during a crime scene investigation and special care is needed when collecting forensic samples [3]. In forensic cases, the presence or absence of specific mites may provide clues [7] and evidence which may be admissible in court [19,20,21,22]. For these and other reasons, accurate mite identification is essential as a potential indicator of habitat, geographic localization, and time markers.

The aim of this paper is to contribute to the knowledge of phoretic Gamasina mites associated with animal remains in a particular Mediterranean region. A thorough knowledge of specific adaptations of mites to ephemeral environments is needed in order to better apply the information provided by mites to the results of forensic investigations.

## 2. Materials and Methods

The fieldwork was carried out in an enclosed triangular area of 600 m^2^ ((40 m × 30)/2) designated for scientific research in “El Carrascal” (Unzué) (42.651411, −1.641214), located in central Navarra (northern Spain). “El Carrascal” is a Mediterranean forest located at an altitude of 568 m.a.s.l. resulting from the degradation of a natural forest of *Quercus rotundifolia* Lam. in which *Quercus coccifera* L. currently predominates. Following Köppen’s classification [23], the weather in this zone is Mediterranean with cool summers. In spring (March, April, and May), the average temperature reaches 13.5–15.5 °C. The accumulated precipitation is about 125–250 L/m^2^ and average daily insolation is 5.7–8.5 h depending on the month and zone. Summers (June, July, and August) are hot, sunny, and without rain. The average temperature is 20–22 °C, average precipitation is 90–125 L/m^2^, and the average insolation is 9–10 h. It is close to a highway and 1.5 km from the nearest urban center (in a straight line). The location is relatively isolated with limited human disturbances or problems related to bad odors, yet it is not far from a major road. As such, this area shares similar characteristics to places where cadavers are likely to be found [24,25,26].

Faunistic samples were taken using three types of traps: ‘modified McPhail’ (*D*) (Figure 1), placed approximately 2 m above the ground; ‘modified pitfall’ (*C*) (Figure 2); and ‘carrion on surface’ (*L*) (Figure 3). The *D* trap was an inverted McPhail trap, with a plate-like structure added inside to place c.a. 20 g of bait and 30% propylene glycol as preservative. The *C* trap was a pitfall trap with a small amount of propylene glycol and c.a. 20 g of bait hanging from a metal structure placed above the trap, and *L* traps consisted of c.a. 150 g of bait placed directly on the ground with four jars to entrap mesofauna around it. The *D* traps attracted and captured flying insects and *C* and *L* traps attracted flying and walking insects. The design of the *C* and *D* traps allowed us to catch mites in their phoretic stage and the aim of the *L* traps was to catch adult mites whose phoretic deutonymphs could not be determined to species level (this trap has the advantage of allowing mites to continue to develop). Every trap (except *D*) was enclosed within metal cages to protect them from wild animals and the bait used was pig (*Sus scrofa domesticus*) entrails (heart, lungs, and liver), a model accepted as representative of human decomposition [27].

When a mite specimen detached and fell into the collecting medium, noting the type of trap where the mite was captured enabled us to deduce if it had arrived in the traps by way of a flying or walking insect.

In this study, a total of 224 traps were analyzed: 96 *D*, 96 *C*, and 32 *L*. During each sampling week 8 *D*, 8 *C*, and 2 *L* traps were set. The relative position of the traps and the distance between them is shown in Figure 4.

The traps remained in the field for approximately one week before being removed once the bait became dry. The sampling periods extended from 7 April, when the traps were placed, to 24 June 2017. The collection dates were as follows: 11 April, 24 April, 29 April, 7 May, 13 May, 21 May, 27 May, 4 June, 10 June, 17 June, and 24 June.

The contents of each trap were preserved in glass jars with 70% ethanol to avoid fungus and bacterial proliferation, which could potentially damage the samples while awaiting analysis. In the laboratory, mites in the preservative liquid were separated from other necrophiles/necrophagous arthropods, counted according to the operational taxonomic unit (OTU), then stored in individual vials in 70% alcohol. Finally, some individuals of each OTU were cleared using Nesbitt’s fluid, mounted in Hoyer’s medium on microscope slides, and identified to species level (if possible) using a compound microscope (OLYMPUS OPTICAL CO., LTD. model BX51TF, Japan) equipped with a phase contrast optical system. Identification was made following specialized taxonomic keys [1,2,5,28,29,30,31,32,33,34].

Table 1 and Table 2 detail data about captured Gamasina species following the order proposed by Hallan [35] listed together with other related biological aspects. Females without spermatophore inside their accessory sperm system were considered as virgin (not having mated) females. Mated females bear spermatophores, eggs, or larvae inside their bodies.

Specimens are deposited in the Museum of Zoology, University of Navarra (MZUNAV), Pamplona, Spain.

## 3. Results

Of the 224 traps analyzed, 15% of *D*, 39% of *C*, and 28% of *L* traps contained mites, meaning that trapping efficiency was 27%. However, 100% of every type contained insects.

Altogether, 844 gamasid mites were collected from traps baited with decomposing organic matter. They belonged to 7 families, 15 genera, and 26 species (Table 1).

The most diverse families were Parasitidae, represented by 11 species, followed by Macrochelidae and Eviphididae, represented by 4 species for each family.

The most abundant species were *Poecilochirus subterraneus* (268 specimens), *Macrocheles merdarius* (116), *Poecilochirus austroasiaticus* (111), and *Poecilochirus carabi* (91). Fewer than 50 individuals were collected from the other species (Table 1).

Adult females were collected only from families Macrochelidae, Ascidae, Veigaiidae, Laelapidae, in the genus *Alliphis*, and in the species *Parasitus (Coleogamasus) americanus, Pergamasus* sp., and *Scarabaspis inexpectatus*. For the species *Macrocheles glaber*, *Macrocheles merdarius*, *Macrocheles muscaedomesticae*, and *Cosmolaelaps lutegiensis*, some females carried an egg or a larva internally. For the other species, deutonymphs were collected and only two males of the family Parasitidae were found. Its presence will be discussed later.

In *D* traps, the species *Gamasodes spiniger* and *Macrocheles muscaedomesticae* were collected, both associated with flying insects. *Parasitus (Coleogamasus) americanus*, *Pergamasus* sp., *Vulgarogamasus remberti*, *Alliphis kargi*, *Halolaelaps* sp., *Cosmolaelaps vacua* (Michael, 1891), *Hypoaspis (Geolaelaps)* sp., and *Cosmolaelaps lutegiensis* (Shcherbak, 1971) were found exclusively in *C* traps. *Parasitus consanguineus* Oudemans & Voigts, 1904, *Veigaia planicola* (Berlese, 1892), *Glyptholaspis confusa*, *Macrocheles glaber*, *Alliphis necrophilus*, *Scarabaspis inexpectatus*, and *Proctolaelaps* sp. were found exclusively in *L* traps; and *Cornigamasus lunaris*, *Parasitus coleoptratorum*, *Parasitus fimetorum*, *Poecilochirus austroasiaticus* Vitzthum, 1930, *Poecilochirus carabi* G. & R. Canestrini, 1882, *Poecilochirus subterraneus*, *Macrocheles merdarius*, *Crassicheles holsaticus*, and *Halolaelaps octoclavatus* were found in both *C* and *L* traps (Table 1). The most effective trap, in terms of the diversity of what was captured, was the pitfall trap (*C*). The trap captured 19 mite species from the 26 reported in this research.

During the three-month monitoring period, a progressive increase in the number of species was observed (Table 2), with 2 species being the average number found in April, 5 species in May, and 13 species in June.

Regarding the presence/absence of the species, three different patterns were observed: species such as *P. coleoptratorum*, *P. austroasiaticus*, *P. carabi*, and *P. subterraneus* appeared continually from their first appearance (2 or 3 weeks after placement of traps) until 24 June; *G. spiniger*, *P. fimetorum*, *M. muscaedomesticae*, and *C. holsaticus* were present only intermittently; and species such as *V. remberti*, *G. confusa*, and *Halolaelaps* sp. were occasional (appearing only on one occasion 5 weeks after the placement of traps). *M. muscaedomesticae* arrived the first collection day, 1 week after the placement of traps (11 April). *Poecilochirus* species delayed their presence, with *P. austroasiaticus* (fifth week, 13 May) arriving before *P. carabi* and *P*. *subterraneus*, which were collected 3 weeks later (4 June).

A total of 16 species are new records for the Iberian Peninsula: *Cornigamasus lunaris* (Berlese, 1882), *Gamasodes spiniger* (Trägårdh, 1910), *Parasitus (Coleogamasus) americanus* (Berlese, 1906), *Parasitus coleoptratorum* (Linneaus, 1758), *Parasitus fimetorum* (Berlese, 1903), *Poecilochirus subterraneus* (Müller, 1859), *Vulgarogamasus remberti* (Oudemans, 1912), *Glyptholaspis confusa* (Foá, 1900), *Macrocheles glaber* (Müller, 1860), *Macrocheles merdarius* (Berlese, 1889), *Alliphis kargi* Arutunian, 1991, *Alliphis necrophilus* Christie, 1983, *Crassicheles holsaticus* Willm, 1937, *Scarabaspis inexpectatus* (Oudemans, 1903), *Halolaelaps octoclavatus* (Vitzthum, 1920), and *Cosmolaelaps lutegiensis* (Shcherbak, 1971).

## 4. Discussion

Species in the families Parasitidae, Macrochelidae, Ascidae, Laelapidae, and Dinychidae (Uropodina) are frequently found on human corpses [7,36,37], reaching them via phoresy on insects. Each species colonizes a corpse at a specific phoretic development stage, the most common being the virgin and gravid females and the deutonymphs of both sexes [6]. Males rarely travel phoretically. The accurate determination of phoretic stage and the correct identification of the species are both critical. Knowing the evolutionarily selected phoretic stage of a given species and the development time of each stage of its lifecycle, an expert could accurately calculate the time elapsed from the moment of colonization to the moment in which that species is found [38]. For example, Macrochelidae only travel as adult females [39], whereas Parasitidae has selected deutonymphs of both sexes as the dispersive stage [1]. In other families, as for example *Scarabaspis inexpectatus* (Eviphididae), adult females and males as well as deutonymphs of both sexes may be the phoretic travelers [40].

Phoresy is common in Macrochelidae and their main hosts are Diptera and Coleoptera [2,41], with high specificity in terms of host preferences. Species of *Glyptholaspis* are carried by scarab beetles and muscid flies [1,7,42]; *Macrocheles glaber* has been found as a phoretic mite on beetles of the families Scarabaeidae, Staphylinidae, Carabidae, and Histeridae [40,43], on genus *Nicrophorus* (Silphidae), and on Muscidae [1,2,42,44]. *Macrocheles merdarius* is rarely found on flies; it is an opportunistic species on scarab beetles [1,40,42,45,46]. *Macrocheles muscaedomesticae* is a specialist species of flies [42]. It is frequently found on *Musca domestica* [1,41,44,46,47], but it has also been reported as phoretic on *Fannia* spp., *Stomoxys calcitrans* Linnaeus, 1758, and *Drosophila simulans* Sturtevant, 1919 [2,48]. *M. muscaedomesticae* may occasionally be transported by beetles of the genus *Penton* and *Geotrupes* [40].

Phoretic *Macrocheles* species always select the female, although the condition in which this female travels may vary: some are virgin females and others are mated females. We were sure that mated females travel because they carried eggs or larvae in their bodies and no males traveled with them, nor were they waiting for them in the trap to mate with them (males were not found in the traps). This phenomenon was previously observed and commented upon [40,49,50,51], the conclusion being that during phoretic travel, mite embryos continue their development while awaiting an optimum environment. However, Filiponni et al. explained that this reproductive behavior is influenced by the quality of the substrate remaining at the previous habitat and by the mite feeding prior to travel [50].

Some members of the family Parasitidae are adapted to ephemeral environments and have developed a phoretic strategy to disperse, selecting the deutonymph as the traveling stage [1]. Phoretic deutonymphs are associated with flies [52,53] and beetles [1,5,7,10,14,16,31,40,41,45,54,55,56,57,58,59,60,61,62,63], although the grade of specificity with their hosts may vary depending on the species [8]. Adult specimens of *P. coleoptratorum* and *P. (C.) americanus* were found in *L* traps in this study. We suggest that deutonymphs of both species fell into the traps when their host insect came in contact with the bait and laid its eggs. This insect behavior could be the signal that induces deutonymphs to detach from the host’s body [64] and continue their development in the new habitat, subsequently molting into the adult stage.

*P. carabi* and *P. necrophori* had been considered as synonymous based on external anatomy [31,33]. Baker and Schwarz [59] rejected the synonymy because both species are reproductively isolated. Our specimens were identified as *P. carabi* after many morphological considerations. After re-examining specimens previously found in Navarra cited as *P. necrophori* [65], we conclude that those specimens should be assigned to *P. carabi.* As such, we do not consider *P. carabi* as a new record for the Iberian Peninsula. Nevertheless, a deep revision of different populations of both morphotypes is clearly needed to obtain a better understanding of the biology and dynamics of these species.

*Vulgarogamasus remberti* is a species not normally associated with decomposing organic matter, but it is associated with mice and voles [31]. Its presence may be due to shrews, which occasionally fall into the jars.

In the family Eviphididae, phoresy is common on flies and carrion beetles [1,34]. *Alliphis necrophilus* specifically inhabits decaying animal tissue [34] and is often associated with Silphidae [66]. *Alliphis kargi* is phoretic on Scarabaeidae [34] and females, males, and deutonymphs were found on *Scarabaspis inexpectatus* [40]. *Crassicheles holsaticus* is reported associated with Sphaeroceridae [67] and Staphylinidae [30] in Russia and England, respectively.

*Proctolaelaps euserratus* Karg, 1994 was cited in Spain, being the most abundant mite in a single human corpse in August [25,68]. A different unclassified species of *Proctolaelaps* was collected in this study whose low abundance could increase from spring to summer (the last sampling date of this study was 24 June).

*Veigaia planicola* has been previously found in Navarra [65,69]. This species is frequently found in the surface layer of the soil, forest litter, and occasionally in rodent nests [1]. Its presence in the trap may be due to the visits of shrews, which feed on carrion insects, or merely due to the presence of traps on the surface of their natural habitat.

The presence or absence of species in a given location depends on the season, the geographical region [36], and the abundance and dispersal strategy of their hosts [42]. This research started in spring (11 April) and finished in early summer (24 June); the increase in the average number of species present each month could be related to the increase in temperature and, as a consequence, an increase in diversity and activity of the available hosts.

## 5. Conclusions

Mites are associated with animal remains and exhibit high specificity with regard to their insect carriers. Therefore, they should not be ignored as a forensic tool and special care should be taken when processing insects collected from a crime scene or from other forensic evidence, as they may carry mites attached to their bodies or detached in the preservative fluid. Accordingly, it is necessary to have a deep understanding of the diversity present in a given region beforehand, together with knowledge of the biology and ecology of each mite species and its insect host. Moreover, it is necessary to be especially careful with those species known to have more than one phoretic stage, and with species with more than one phenotype. An incorrect identification could introduce errors in the interpretation of the data.

The most abundant species in this research was *Poecilochirus subterraneus* (deutonymphs) (traps *C* and *L*), *Poecilochirus austroasiaticus* (traps *C* and *L*), and *Poecilochirus carabi* (traps *C* and *L*), and virgin and mated females of *Macrocheles merdarius* (traps *C* and *L*). *Gamasodes spiniger* was the only species collected in all three types of traps (*C*, *D*, and *L*). These five species of mites may be able to provide crucial information to an expert when examining forensic organic remains. During the spring in the Mediterranean region of Navarra (northern Spain), phoretic mites on carrion insects exhibited three patterns regarding presence/absence: occasional, intermittent, and continual.

Of the 26 species reported in this research, 7 species of the family Parasitidae, 3 species of Macrochelidae, 4 species of Eviphididae, 1 species of Halolaelapidae, and 1 species of Laelapidae are new records for the Iberian Peninsula.

## Figures and Tables

**Figure 1 insects-10-00005-f001:**
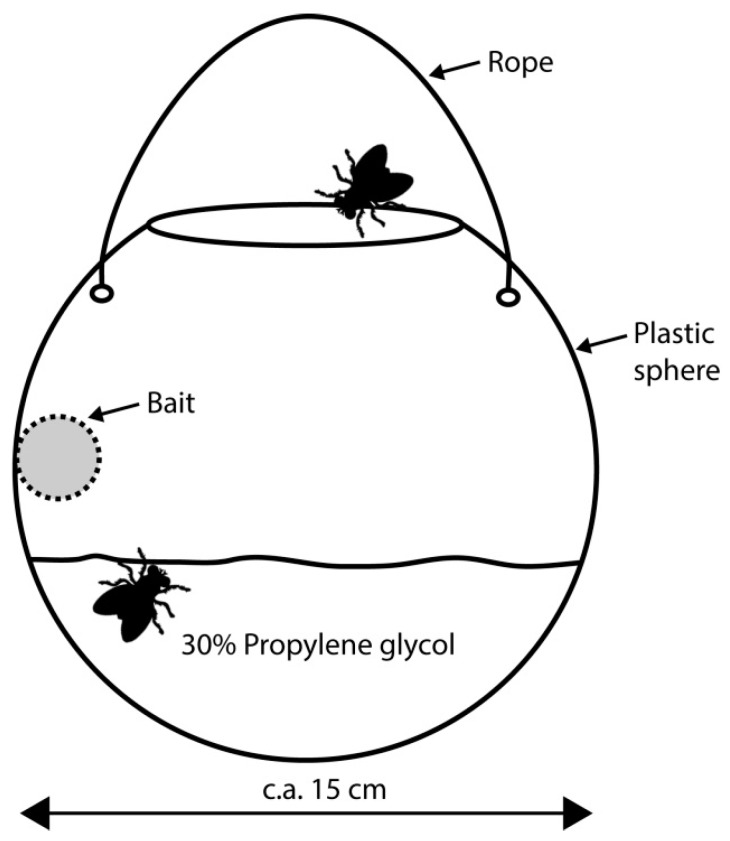
Modified McPhail trap (*D*).

**Figure 2 insects-10-00005-f002:**
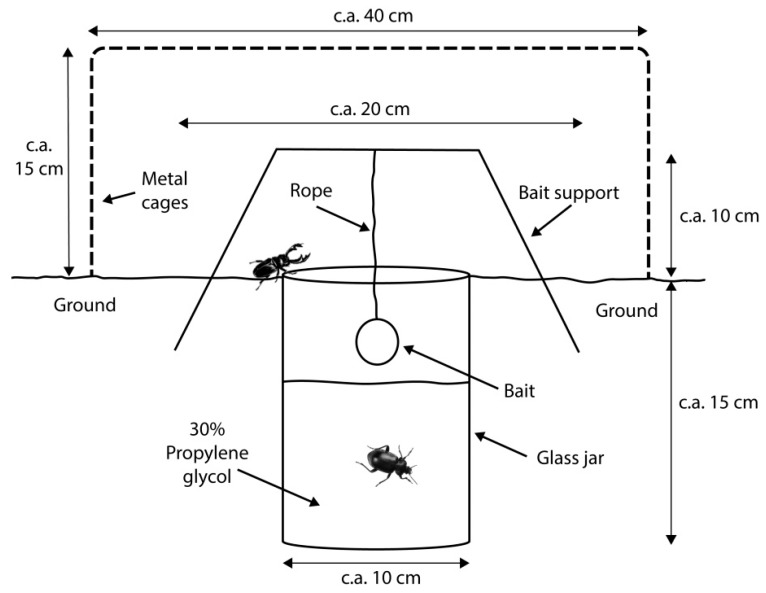
Modified pitfall trap (*C*).

**Figure 3 insects-10-00005-f003:**
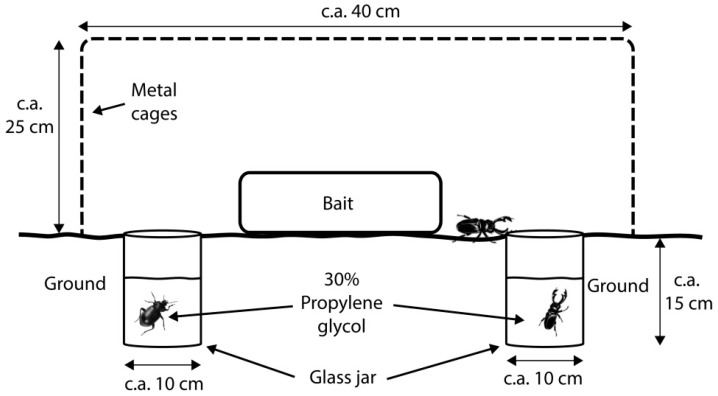
Carrion on surface trap (*L*).

**Figure 4 insects-10-00005-f004:**
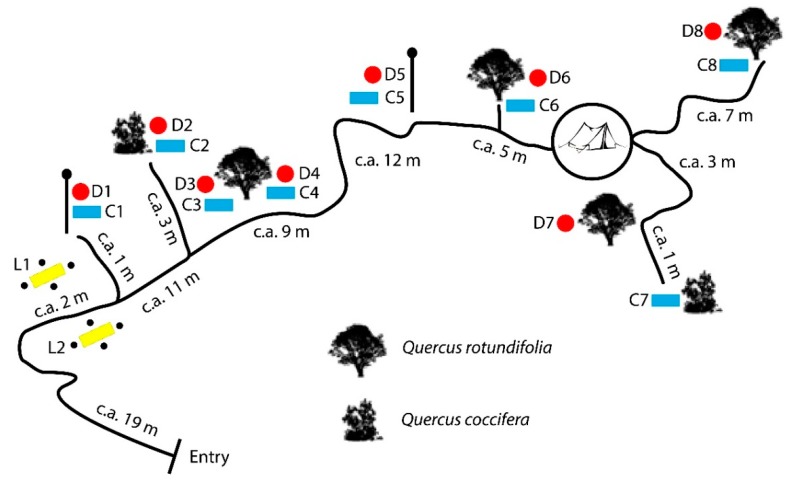
The location of the traps in the research field. Black lines represent the path connecting traps; solid red circles represent McPhail traps (*D*); solid blue rectangles represent pitfall traps (*C*); solid yellow rectangles represent carrion on surface (*L*) traps; small solid black circles represent jars.

**Table 1 insects-10-00005-t001:** Mites collected from April to June (2017).

Sp. No.	Family	Species	No.	Stage	Trap
1	Parasitidae	*Cornigamasus lunaris* (Berlese, 1882)	2	DN	*C*, *L*
2		*Gamasodes spiniger* (Trägårdh, 1910)	34	DN	*C*, *D*, *L*
3		*Parasitus (Coleogamasus) americanus* (Berlese, 1906)	1	♀ e	*C*
4		*Parasitus coleoptratorum* (Linneaus, 1758)	38	♂, DN	*C*, *L*
5		*Parasitus consanguineus* Oudemans & Voigts, 1904	2	DN	*L*
6		*Parasitus fimetorum* (Berlese, 1903)	20	DN	*C*, *L*
7		*Pergamasus* sp.	2	♀, ♂	*C*
8		*Poecilochirus austroasiaticus* Vitzthum, 1930	111	DN	*C*, *L*
9		*Poecilochirus carabi* G. & R. Canestrini, 1882	91	DN	*C*, *L*
10		*Poecilochirus subterraneus* (Müller, 1859)	268	DN	*C*, *L*
11		*Vulgarogamasus remberti* (Oudemans, 1912)	1	DN	*C*
12	Veigaiidae	*Veigaia planicola* (Berlese, 1892)	1	♀	*L*
13	Macrochelidae	*Glyptholaspis confusa* (Foá, 1900)	1	♀	*L*
14		*Macrocheles glaber* (Müller, 1860)	45	♀, ♀ e, ♀ Lv	*L*
15		*Macrocheles merdarius* (Berlese, 1889)	116	♀, ♀ e, ♀ Lv	*C*, *L*
16		*Macrocheles muscaedomesticae* (Scopoli, 1772)	27	♀ e, ♀ Lv	*D*, *C*
17	Eviphididae	*Alliphis kargi* Arutunian, 1991	1	♀	*C*
18		*Alliphis necrophilus* Christie, 1983	1	♀	*L*
19		*Crassicheles holsaticus* Willm, 1937	21	DN	*C*, *L*
20		*Scarabaspis inexpectatus* (Oudemans, 1903)	3	♀, DN	*L*
21	Ascidae	*Proctolaelaps* sp.	7	♀	*L*
22	Halolaelapidae	*Halolaelaps* sp.	1	DN	*C*
23		*Halolaelaps octoclavatus* (Vitzthum, 1920)	34	DN	*C*, *L*
24	Laelapidae	*Cosmolaelaps vacua* (Michael, 1891)	2	♀	*C*
25		*Cosmolaelaps lutegiensis* (Shcherbak, 1971)	12	♀, ♀ e	*C*
26		*Hypoaspis (Gaeolaelaps)* sp.	2	♀	*C*
		Total collected:	844		

List of families and species collected. 1–26 Numbers correspond to species in the third column (Sp. No.); number of specimens (No.); stage and sex (Stage) of the captured specimens are abbreviated as follows: gravid female with an egg and larva inside (♀ Lv), gravid female with egg (♀ e), female (♀), male (♂) and deutonymph (DN); type of traps (Trap) in which they were found.

**Table 2 insects-10-00005-t002:** Table of presence/absence of species found in each sampling.

Sp. No.	11 April	24 April	29 April	7 May	13 May	21 May	27 May	4 June	10 June	17 June	24 June
1											•
2		•		•	•		•	•	•	•	•
3					•						
4							•	•	•	•	•
5										•	•
6				•	•				•	•	•
7											
8											
9											
10											
11											
12											
13											
14											
15											
16											
17											
18											
19											
20											
21											
22											
23											
24											
25											
26											

Presence (black cells)/absence (white cells) of collected species in each sampling date. Numbers 1–26 in the first column (Sp. No.) correspond with the species names in Table 1.

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
