# Peer review of "Gamasina Mites (Acari: Mesostigmata) Associated with Animal Remains in the Mediterranean Region of Navarra (Northern Spain)"

_insects, 2019, doi:10.3390/insects10010005_

Round 1

Reviewer 1 Report

The authors surveyed mites attracted to pork-bated traps in north central Spain.  Their justification for doing the work was to promote the use of mites in death investigation.  Mites are indeed commonly associated with a corpse, and few scientists have the specialized knowledge needed to identify them, so these results should be published.  However, revision of the manuscript is required.  In particular, the authors need to concentrate on forensically useful details, and much less on the general biology of the mites.

The introduction and discussion include a detailed review of the phenomenon of phoresy by mites, including evolution. This material has almost no forensic relevance, and only a few sentences on the topic are needed.  A mite may colonize a corpse on its own, or by phoresy on a carrion insect. A phoretic species could be represented by a larger number of life stages.  What else does a forensic entomologist need to know?

The methods are difficult to understand. Photographs or illustrations of the traps in place would help.  The authors emphasize the habitat-specificity of mites, and therefore of the potential utility of mites for reconstructing the relocation of a corpse, but they did not describe the habitat(s) of the trap location(s), distance between traps, etc., other than to say it was isolated.

For succession analysis a valuable species is one found on almost every corpse (see Perez et al. 2014. Forensic Science International 241:91-95). The authors should say if the abundant species were also collected in every trap of a given type during each trapping period.

It is okay to publish preliminary results, but the authors should mention that succession studies with intact carcasses must be done before it will be clear that these mites can be used to estimate PMI.

Lines 63-64: Why in the absence of insects?  Does the interpretation of mite data depend on whether or not insects are present?

Line 107: What are the other specific papers? Please provide voucher numbers for specimens, at least for the new distribution records.

Lines 169-170: Please say more.  Do the authors mean that the different life stages are found during different PMI values?  Also, what do they mean by time of colonization?  Colonization by that individual mite? Colonization by the initial carrion arthropod?  A succession pattern includes the time period when there are not yet any carrion arthropods, so succession analysis gives you PMI, not a minimum PMI (see Villet and Amendt 2011. Advances in entomological methods for death time estimation. In Forensic Pathology Reviews volume 6).

Lines 259-261.  Perhaps funding from the Association of Friends of the University of Navarra is not "external," but the authors should confirm that this is what the journal means by "external."

Author Response

“Their justification for doing the work was to promote the use of mites in death investigation”. Our final justification for doing this research was to promote the use of mites in forensic matters not only in death investigations. The main aim is to increase the knowledgement of phoretic mites and their hosts in a Mediterranean region.

“What else does a forensic entomologist need to know? A forensic entomologist needs to be able to do accurate identification of the mite species and be aware about all the biology and ecological matters of mites of forensic importance.

“The methods are difficult to understand”. Description of the methodology has been improved. A map of the research field with traps locations has been included together with figures of the three types of traps.

“Photographs or illustrations of the traps in place would help”. A map of the research field with traps locations has been included together with figures of the three types of traps.

“The authors emphasize the habitat-specificity of mites, and therefore of the potential utility of mites for reconstructing the relocation of a corpse, but they did not describe the habitat(s) of the trap location(s), distance between traps, etc., other than to say it was isolated”. The potential utility of mites for reconstructing the relocation of a corpse was the aim of this paper. Several sentences about the habitat, a map with distance between traps, etc…, and other figures were included.

For succession analysis a valuable species is one found on almost every corpse (see Perez et al. 2014. Forensic Science International 241:91-95). The authors should say if the abundant species were also collected in every trap of a given type during each trapping period. Table 1 gives information about every species and the type of trap where it was collected during the sampling period. Other paragraphs include the information requested.

It is okay to publish preliminary results, but the authors should mention that succession studies with intact carcasses must be done before it will be clear that these mites can be used to estimate PMI. This research was not designed to estimate PMI in death cases.

Lines 63-64: Why in the absence of insects? Does the interpretation of mite data depend on whether or not insects are present? The absence of a given species in a given moment of organic putrefaction or in the presence of certain chemical molecules, can act as a repellent of certain taxa.

Line 107: What are the other specific papers? We delete the sentence since no other papers were used to identify the species.

Please provide voucher numbers for specimens, at least for the new distribution records. Voucher numbers will be assigned for each specimen when those are officially incorporated into the MZUNAV at the end of this research.

Lines 169-170: Please say more. Do the authors mean that the different life stages are found during different PMI values? Every sentence directly related with PMI concept have been delete.

Also, what do they mean by time of colonization? Colonization by that individual mite? Colonization by the initial carrion arthropod?  Time elapsed from the moment the mite species arrived to the substrate to the moment the species is found.

A succession pattern includes the time period when there are not yet any carrion arthropods, so succession analysis gives you PMI, not a minimum PMI (see Villet and Amendt 2011. Advances in entomological methods for death time estimation. In Forensic Pathology Reviews volume 6). Every sentence directly related with PMI concept have been delete

Lines 259-261. Perhaps funding from the Association of Friends of the University of Navarra is not "external," but the authors should confirm that this is what the journal means by "external." This grant is not “external”. It is a grant for PhD students.

Reviewer 2 Report

Please see my edits, comments and suggestions on the attached pdf file.

I realize that some of the problems in the manuscript are due to the fact that English may not be the first language of the authors. Those problems can easily be fixed, I think, with a good translator. 

My biggest concern with this manuscript is that the authors claim that a "moment of death" can be determined from the absence/presence of mites found on a corpse. They presented no scientific data to back up such a claim.   

I do agree that in some cases, mites may be used as tools in forensic entomology, and the authors have done a good job of documenting which species are associated with carrion in Navarra, Spain, during different seasons. But this is far different than determination of the moment of death. 

If you take out the "moment of death" sentences, the manuscript has informative data and needs only some corrections in English spelling, grammar and sentence structure.          

I would also like to see a little more basic information about the lifecycle mites. 

Author Response

Please see my edits, comments and suggestions on the attached pdf file. We have taking into account every comment and correction made in the pdf of the manuscript. Several sentences have been rewritten and new sentences have been added.

The manuscript has been check for two native English speakers.

My biggest concern with this manuscript is that the authors claim that a "moment of death" can be determined from the absence/presence of mites found on a corpse. They presented no scientific data to back up such a claim. All sentences directly concerning “moment of death” have been delete since it is not the main aim of this paper.

I do agree that in some cases, mites may be used as tools in forensic entomology, and the authors have done a good job of documenting which species are associated with carrion in Navarra, Spain, during different seasons. But this is far different than determination of the moment of death. We know that and sentences referring this have been deleted.

If you take out the "moment of death" sentences, the manuscript has informative data and needs only some corrections in English spelling, grammar and sentence structure. It have been done. The manuscript has been check for two native English speakers.

I would also like to see a little more basic information about the lifecycle mites. Basic information about reproduction and lifecycle has been inserted.

Reviewer 3 Report

This is a nice paper.

Abstract

— Please include more actual results in the abstract. Please be more specific in the abstract.

— E.g., he "denomination" of the name of pitfall trap does not belong in the abstract.

General 

Language must be improved by a native speaker; e.g. "shrewmice" = shrew mice but mostly because:

Examples of content that I did not understand (but I guess just due to grammar / style):

— there exists polymorphism

— originating preference

— to establish timeline

— The fieldwork was an enclosed area

— to deep on knowledge

= please be more clear / use better english terms.

Please unify "grams" vs. "g."

Please check commas.

line 226: what exactly do you mean by "removal"? Please explain in paper.

Discussion

— Please discuss if mites will be transported by flies to the scene of crime very early. You already mention that mites might arrive early but how early and when and how faster that by distribution by flies?

— Since this is an in insect journal, pls make a reference to insecs on corpses, e.g. https://pdfs.semanticscholar.org/044e/31ac421c0ff706ab9f329ea5e1c6e4f89d98.pdf

Conclusions

Please mention more precisely how your results may be seen in connection to your references 19, 20, 22.

Again, the paper is nice but needs a bit polishing.

Author Response

Abstract

—Please include more actual results in the abstract. Please be more specific in the abstract. More specific information was added. More information has been included but the journal only allow 200 words in this section.

—E.g., The "denomination" of the name of pitfall trap does not belong in the abstract. The journal asks for accurate information and the type of traps used in this paper is important.

General

Language must be improved by a native speaker; e.g. "shrewmice" = shrew mice but mostly because:

The manuscript has been check for two native English speakers.

Examples of content that I did not understand (but I guess just due to grammar / style):

— there exists polymorphism

— originating preference

— to establish timeline

— The fieldwork was an enclosed area

— to deep on knowledge

= please be more clear / use better english terms.

Please unify "grams" vs. "g." Done

Please check commas. Done

The manuscript has been check for two native English speakers. The style has been checked and corrected and several sentences rewritten.

line 226: what exactly do you mean by "removal"? Please explain in paper. “Removal” was deleted in the manuscript.

Discussion

— Please discuss if mites will be transported by flies to the scene of crime very early. You already mention that mites might arrive early but how early and when and how faster that by distribution by flies? All sentences concerning death has been removed since it is not the main aim of the paper. We are not doing a successional study. Some of this information could be read in the references of this manuscript.

— Since this is an in insect journal, pls make a reference to insecs on corpses, e.g. https://pdfs.semanticscholar.org/044e/31ac421c0ff706ab9f329ea5e1c6e4f89d98.pdf

Data about insect hosts of these mites will be the aim of a future paper. The paper you suggested is a brief history of forensic entomology and about insect and human corpses. There are many references about insect hosts.

Conclusions

Please mention more precisely how your results may be seen in connection to your references 19, 20, 22.

The “death” was not the main aim of this paper although the results given will be applied in future research about PMI and other matters about legal forensic entomology.

Round 2

Reviewer 3 Report

Just one final thing: Please change "Mites cannot be overlooked" to "Mites should not be...". Apart from that, the redo of the paper is fine. Thank you.

Author Response

Just one final thing: Please change "Mites cannot be overlooked" to "Mites should not be...". Apart from that, the redo of the paper is fine. Thank you.

The change was done.